# Differentiable Approximations for Multi-resource Spatial Coverage Problems

## Abstract

Resource allocation for coverage of physical spaces is a challenging problem in robotic surveillance, mobile sensor networks and security domains. Recent gradient-based optimization approaches to this problem estimate utilities of actions by using neural networks to learn a differentiable approximation to spatial coverage objectives. In this work, we empirically show that spatial coverage objectives with multiple-resources are combinatorially hard to approximate for neural networks and lead to sub-optimal policies. As our major contribution, we propose a tractable framework to approximate a general class of spatial coverage objectives and their gradients using a combination of Newton-Leibniz theorem, spatial discretization and implicit boundary differentiation. We empirically demonstrate the efficacy of our proposed framework on single and multi-agent spatial coverage problems.

## 1 Introduction

Allocation of multiple resources for efficient spatial coverage is an important component of many practical single-agent and multi-agent systems, for e.g., robotic surveillance, mobile sensor networks and security game modeling. Surveillance tasks generally involve a single agent assigning resources e.g. drones or sensors, each of which can monitor physical areas, to various points in a target domain such that a loss function associated with coverage of the domain is minimized (Renzaglia et al., 2012). Alternatively, security domains follow a leader-follower game setup between two agents, where a defender defends a set of targets (or a continuous target density in a geographical area) with limited resources to be placed, while an attacker plans an attack after observing the defender's placement strategy using its own resources (Tambe, 2011).

Traditional methods used to solve *single-agent multi-resource* surveillance problems often rely on potential fields (Howard et al., 2002), discretization based approaches (Kong et al., 2006), voronoi tessellations (Dirafzoon et al., 2011) and particle swarm optimization (Nazif et al., 2010; Saska et al., 2014). Similarly, many exact and approximate approaches have been proposed to maximize the defender's expected utility in *two-agent multi-resource* security domains against a best responding attacker (Kiekintveld et al., 2009; Amin et al., 2016; Yang et al., 2014; Haskell et al., 2014; Johnson et al., 2012; Huang et al., 2020). Notably, most existing traditional approaches focus on exploiting some specific spatio-temporal or symmetry structure of the domain being examined.

**Related Work**: Since spatial coverage problems feature continuous action spaces, a common technique used across most previous works is to discretize the area to be covered into grid cells and restrict the agents' actions to discrete sets (Kong et al., 2006; Yang et al., 2014; Haskell et al., 2014; Gan et al., 2017) to find the equilibrium mixed strategies or optimal pure strategies using integer linear programming. However, discretization quickly becomes intractable when the number of each agent's resources grows large. While some games can be characterized by succinct agent strategies and can be solved efficiently via mathematical programming after discretizing the agents' actions spaces (Behnezhad et al., 2018), this is not true for most multi-resource games.

Recent works in spatial coverage domains have focused on incorporating advances from deep learning to solve the coverage problems with more general algorithms. For instance, Pham et al. (2018) focus on the multi-UAV coverage of a field of interest using a model-free multi-agent RL method while StackGrad (Amin et al., 2016), OptGradFP (Kamra et al., 2018), PSRO (Lanctot et al., 2017) are model-free fictitious play based algorithms which can be used to solve games in continuous action spaces. However model-free approaches are sample inefficient and require many interactions with the domain (or with a simulator) to infer expected utilities of agents' actions. Secondly, they often rely

on the policy gradients to compute the derivative of the agents' expected utilities w.r.t. their mixed strategies, which induces a high variance in the estimate.

To alleviate these issues, more recent works take an actor-critic based approach (Lowe et al., 2017), which additionally learns a differentiable approximation to the agents' utilities (Kamra et al., 2019a; Wang et al., 2019) and calculate gradients of strategies w.r.t. the utilities. But this requires learning accurate reward/value functions which becomes combinatorially hard for multi-resource coverage.

**Contributions**: To address the above challenge, we present a framework to tractably approximate a general class of spatial coverage objectives and their gradients via spatial discretization without having to learn neural network based reward models. We only discretize the target domain to represent integrals and all set operations over it, but not the action spaces of the agents. Hence we mitigate the intractability caused by discretizing high dimensional action spaces of agents with large number of resources, while also keeping agents' actions amenable to gradient-based optimization. By combining our framework with existing solution methods, we successfully solve both single-agent and adversarial two-agent multi-resource spatial coverage problems.

## 2 MULTI-RESOURCE SPATIAL COVERAGE PROBLEMS

In this section, we formally introduce notation and definitions for multi-resource allocation problems along with two example applications, which will be used for evaluation.

**Multi-agent multi-resource spatial coverage**: Spatial coverage problems comprise of a target space $Q \subset \mathbb{R}^d$ (generally $d \in \{2, 3\}$) and a set of agents (or players) $P$ with each agent $p \in P$ having $m_p$ resources. We will use the notation $-p$ to denote all agents except $p$ i.e. $P \backslash \{p\}$. **Actions**: An action $u_p \in \mathbb{R}^{m_p \times d_p}$ for agent $p$ is the placement of all its resources in an appropriate coordinate system of dimension $d_p$. Let $U_p$ denote the compact, continuous and convex action set of agent $p$. **Mixed strategies**: We represent a mixed strategy i.e. the probability density of agent $p$ over its action set $U_p$ as $\sigma_p(u_p) \geq 0$ s.t. $\int_{U_p} \sigma_p(u_p) du_p = 1$. We denote agent $p$ sampling an action $u_p \in U_p$ from his mixed strategy density as $u_p \sim \sigma_p$. **Joints**: Joint actions, action sets and densities for all agents together are represented as $u = \{u_p\}_{p \in P}, U = \times_{p \in P} \{U_p\}$ and $\sigma = \{\sigma_p\}_{p \in P}$ respectively. **Coverage**: When placed, each resource covers (often probabilistically) some part of the target space $Q$. Let $\mathrm{cvg}_p : q \times u \to \mathbb{R}$ be a function denoting the utility for agent $p$ coming from a target point $q \in Q$ due to a joint action $u$ for all agents. We do not assume a specific form for the coverage utility $\mathrm{cvg}_p$ and leave it to be defined flexibly, to allow many different coverage applications to be amenable to our framework. **Rewards**: Due to the joint action $u$, each player achieves a coverage reward $r_p : u \to \mathbb{R}$ of the form $r_p(u) = \int_Q \mathrm{cvg}_p(q, u) \mathrm{imp}_p(q) \, dq$, where $\mathrm{imp}_p(q)$ denotes the importance of the target point $q$ for agent $p$. With a joint mixed strategy $\sigma$, player $p$ achieves expected utility: $\mathbb{E}_{u \sim \sigma}[r_p] = \int_U r_p(u) \sigma(u) du$. **Objectives**: In single-agent settings, the agent would directly optimize his expected utility w.r.t. action $u_p$. But in multi-agent settings, the expected utilities of agents depend on other agents' actions and hence cannot be maximized with a deterministic resource allocation due to potential exploitation by other agents. Instead agents aim to achieve Nash equilibrium mixed strategies $\sigma = \{\sigma_p\}_{p \in P}$ over their action spaces. **Nash equilibria**: A joint mixed strategy $\sigma^* = \{\sigma_p^*\}_{p \in P}$ is said to be a Nash equilibrium if no agent can increase its expected utility by changing its strategy while the other agents stick to their current strategy.

**Two-player settings**: While our proposed framework is not restricted to the number of agents or utility structure of the game, we will focus on single-player settings and zero-sum two-player games in subsequent examples. An additional concept required by fictitious play in two-player settings is that of a best response. A best response of agent $p$ against strategy $\sigma_{-p}$ is an action which maximizes his expected utility against $\sigma_{-p}$:

$$ br_p(\sigma_{-p}) \in \arg \max_{u_p} \left\{ \mathbb{E}_{u_{-p} \sim \sigma_{-p}} [r_p(u_p, u_{-p})] \right\}. $$

The expected utility of any best response of agent $p$ is called the exploitability of agent $-p$:

$$ \epsilon_{-p}(\sigma_{-p}) := \max_{u_p} \left\{ \mathbb{E}_{u_{-p} \sim \sigma_{-p}} [r_p(u_p, u_{-p})] \right\}. $$

Notably, a Nash equilibrium mixed strategy for each player is also their least exploitable strategy.

**Example 1** (Single-agent Areal Surveillance). *A single agent, namely the defender (D), allocates $m$ areal drones with the $i^{th}$ drone $D_i$ having three-dimensional coordinates $u_{D,i} = (p_{D,i}, h_{D,i}) \in$*

$[-1,1]^2 \times [0,1]$ *to surveil a two-dimensional forest* $Q \subset [-1,1]^2$ *of arbitrary shape and with a known but arbitrary tree density* $\rho(q)$. *Consequently,* $u_D \in \mathbb{R}^{m \times 3}$. *Each drone has a downward looking camera with a circular lens and with a half-angle* $\theta$ *such that at position* $(p_{D,i}, h_{D,i})$, *the drone* $D_i$ *sees the set of points* $S_{D,i} = \{q \mid ||q - p_{D,i}||_2 \leq h_{D,i} \tan \theta\}$. *A visualization of this problem with* $m = 2$ *drones is shown for a sample forest in Figure 1a. We assume a probabilistic model of coverage with a point* $q$ *being covered by drone* $D_i$ *with probability* $P_H(h_{D,i}) = e^{K(h_{opt} - h_{D,i})} \left( \frac{h_{D,i}}{h_{opt}} \right)^{K h_{opt}}$ *if* $q \in S_{D,i}$ *and* $0$ *otherwise. With multiple drones, the probability of a point* $q$ *being covered can then be written as:* $cvg(q, u_D) = 1 - \prod_{i \mid q \in S_{D,i}} \bar{P}_H(h_{D,i})$ *where* $\bar{P}_H$ *stands for* $1 - P_H$. *Hence, the reward function to be maximized is:* $r_{D,1p}(u_D) = \int_Q \left( 1 - \prod_{i \mid q \in S_{D,i}} \bar{P}_H(h_{D,i}) \right) \rho(q) dq$ *with the tree density* $\rho(q)$ *being the importance of target point* $q$ *(subscript 1p denotes one agent).*

**Example 2** (Two-agent Adversarial Coverage). *Two agents, namely the defender* $D$ *and the attacker* $A$, *compete in a zero-sum game. The defender allocates* $m$ *areal drones with the same coverage model as in example 1. The attacker controls* $n$ *lumberjacks each with ground coordinates* $u_{A,j} \in [-1,1]^2$ *to chop trees in the forest* $Q$. *Consequently,* $u_A \in \mathbb{R}^{n \times 2}$. *Each lumberjack chops a constant fraction* $\kappa$ *of trees in a radius* $R_L$ *around its coordinates* $u_{A,j}$. *We denote the area covered by the* $j$-*th lumberjack as* $S_{A,j} = \{q \mid ||q - p_{A,j}||_2 \leq R_L\}$. *A visualization of this problem with* $m = n = 2$ *is shown for a sample forest in Figure 1b. A drone can potentially catch a lumberjack if its field of view overlaps with the chopping area. For a given resource allocation* $u = (u_D, u_A)$, *we define* $I_j = \{i \mid ||p_{A,j} - p_{D,i}||_2 \leq R_L + h_{D,i} \tan \theta\}$ *as the set of all drones which overlap with the* $j$-*th lumberjack. The areal overlap* $\alpha_{ij} = \int_{S_{D,i} \cap S_{A,j}} dq$ *controls the probability of the* $j$-*th lumberjack being caught by the* $i$-*th drone:* $P_C(h_{D,i}, \alpha_{ij}) = P_H(h_{D,i}) P_A(\alpha_{ij})$ *where* $P_H$ *is the same as that in example 1 and captures the effect of drone's height on quality of coverage, while* $P_A(\alpha_{ij}) = 1 - \exp \left( -\frac{K_a \alpha_{ij}}{\pi R_L^2} \right)$ *captures the effect of areal overlap on probability of being caught. Hence, the reward achieved by the* $j$-*th lumberjack can be computed as:* $r_{A,j}(u_D, u_{A,j}) = \kappa \int_{S_{A,j} \cap Q} \rho(q) dq$ *with probability* $\prod_{i \in I_j} \bar{P}(h_{D,i}, \alpha_{ij})$, *and* $-\kappa \int_{S_{A,j} \cap Q} \rho(q) dq$ *otherwise i.e. the number of trees chopped if the* $j$-*th lumberjack is not caught by any drone or an equivalent negative penalty if it is caught. Hence, the total agent rewards are:* $r_{A,2p}(u_D, u_A) = -r_{D,2p}(u_D, u_A) = \sum_j r_{A,j}(u_D, u_{A,j})$ *(subscript 2p denotes two-agent).*

Note that in the above examples drones provide best probabilistic coverage at a height $h_{opt}$. By increasing their height, a larger area can be covered at the cost of deterioration in coverage probability. Further, the defender can increase coverage probability for regions with high tree density by placing multiple drones to oversee them; in which case, the drones can potentially stay at higher altitudes too. Example 2 further adds additional interactions due to overlaps between defender and attacker's resources[1]. Hence, these examples form a challenging set of evaluation domains with multiple trade-offs and complex possibilities of coverage involving combinatorial interactions between the players' resources. For both examples, we use the following constants: $\theta = \frac{\pi}{6}$, $h_{opt} = 0.2$, $K = 4.0$, $R_L = 0.1$, $K_a = 3.0$, $\kappa = 0.1$. However, note that these values only serve as practical representative values. The techniques that we introduce in this paper are not specific to the above probabilistic capture models or specific values of game constants, but rather apply to a broad class of coverage problems where the agents act by placing resources with finite coverage fields and agents' rewards are of the form: $r_p(u) = \int_Q f_p(u, q) dq$.

**Dealing with zero gradients**: In the two-agent game, the attacker's reward depends on the locations of its resources, but the defender's reward solely depends on overlaps with the attacker's resources. In absence of such overlap, the gradient of $r_{D,2p}$ w.r.t. $u_{D,i}$ becomes 0. Hence, we propose to use the reward from the one-agent game as an intrinsic reward for the defender similar to how RL algorithms employ intrinsic rewards when extrinsic rewards are sparse (Pathak et al., 2017). Then the reward function for the defender becomes: $\tilde{r}_{D,2p}(u_D, u_A) = r_{D,2p}(u_D, u_A) + \mu r_{D,1p}(u_D)$. We use a small $\mu = 0.001$ to not cause significant deviation from the zero-sum structure of the game and yet provide a non-zero gradient to guide the defender's resources in the absence of gradients from $r_{D,2p}$.

---

[1]In reality, lumberjacks might act independent of each other and lack knowledge of each others' plans. By allowing them to be placed via a single attacker and letting them collude, we tackle a more challenging problem and ensure that not all of them get caught by independently going to strongly covered forest regions.

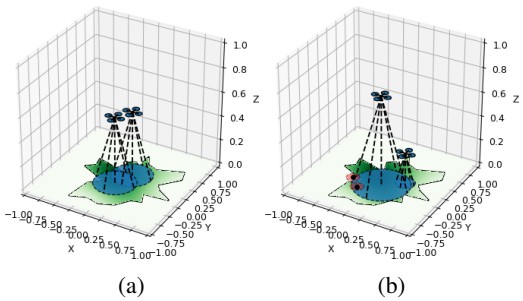

(a)                                      (b)

Figure 1: (a) Areal surveillance example with an arbitrary forest and $m = 2$ drones, (b) Adversarial coverage example with $m = 2$ drones and $n = 2$ lumberjacks (red circles).

## 3 METHODS

**Solution approaches**: The key idea for all solution approaches is to obtain a differentiable approximation to the expected utility of the agents and then maximize it w.r.t. the agents' actions (or mixed strategies). For single-agent games, this boils down to performing direct gradient ascent on a differentiable approximation to $r_D(u_D)$, thereby converging at a (locally) optimal value of $u_D$. For two-agent adversarial games, DeepFP (Kamra et al., 2019b), an actor-critic based approach based on fictitious play can be used. Briefly summarized in algorithm 1, it obtains a differentiable approximation to the reward functions $\tilde{r}_{D,2p}$ and $r_{A,2p}$, creates an empty memory to store a non-parametric representation of the agents' mixed strategies $\sigma = (\sigma_D, \sigma_A)$ and initializes best responses for both agents randomly [lines 1-3]. Then it alternatively updates: (a) the agents' strategies, by storing the current best responses in the memory [line 5], and (b) the best responses, by maximizing each agent $p$'s differentiable reward function against a batch of samples drawn from the other agent's strategy $\sigma_{-p}$ [lines 6-8]. Details of DeepFP hyperparameters used can be found in section A.6 in the appendix. The key component required in both cases is a differentiable approximation to the reward function and we propose a tractable framework for this challenging task in the subsequent sub-sections.

**Mitigating sub-optimal local best responses**: During our preliminary experiments with DeepFP, we observed that updating best responses using purely gradient-based optimization can often get stuck in sub-optimal local optima. While DeepFP maintains stochastic best responses to alleviate this issue, it doesn't eliminate it completely. We briefly describe our solution to this issue here (please see section A.4 in the appendix for a more elaborate discussion on the issue and details of the proposed solution). Motivated by Long et al. (2020), we propose a simple population-based approach wherein we maintain a set of $K$ deterministic best responses $br_p^k(\sigma_{-p})$, for $p \in \{D, A\}$ and $\forall k \in [K]$. During the best response optimization step for agent $p$ [lines 6-8], we optimize the $K$ best responses independently and play the one which exploits agent $-p$ the most. After the optimization step, the top $\frac{K}{2}$ best responses are retained while the bottom half are discarded and freshly initialized with random placements for the next iteration. This allows retention and further refinement of the current best responses over subsequent iterations, while discarding and replacing the ones stuck in sub-optimal local minima.

### 3.1 DIFFERENTIABLE APPROXIMATION FOR COVERAGE OBJECTIVES

First, we propose a method to approximate coverage objectives and their gradients w.r.t. agents' actions. Consider an objective of the form:

$$r(u) = \int_Q f(u, q) \, dq \tag{1}$$

where $u$ denotes actions of one or more agents having multiple resources to place at their disposal and $q$ is any point in the target domain $Q$. We assume that the action $u$ has $m$ components with $u_i$ representing the location of $i$-th resource ($i \in [m]$) and $u_{\setminus i}$ representing the locations of all resources other than $i$. Note that the imp($q$) function has been subsumed into $f(u, q)$ in this formulation. We are interested in computing the gradient: $\frac{\partial r}{\partial u_i}$. However, this is a hard problem since: (a) $r(u)$ involves integration over arbitrary (non-convex shaped) target domains which does not admit a closed-form expression in terms of elementary functions and hence cannot be differentiated with autograd libraries

---

**Algorithm 1:** DeepFP

---

**Result:** Final strategies $\sigma_D, \sigma_A$ in mem

1  Obtain a differentiable approximation $\hat{r} = (\hat{r}_D, \hat{r}_A)$ to the reward functions: $(\tilde{r}_{D,2p}, r_{A,2p})$;

2  Initialize best responses $(br_D, br_A)$ randomly;

3  Create empty memory mem to store $\sigma = (\sigma_D, \sigma_A)$;

4  **for** $game \in \{1, \ldots, max\_games\}$ **do**

  /* Update strategies                 */

5    Update $\sigma$ by storing best responses $\{br_D, br_A\}$ in mem;

  /* Update best responses              */

6    **for** $agent\ p \in \{D, A\}$ **do**

7      Draw samples $\{u_{-p}^i\}_{i=1:bs}$ from $\sigma_{-p}$ in mem;

8      $br_p := \max_{u_p} \frac{1}{bs} \sum_{i=1}^{bs} \hat{r}_p(u_p, u_{-p}^i)$;

---

like PyTorch and TensorFlow, and (b) most resources have a finite coverage area, outside of which the coverage drops to zero. This often makes the function $f(u, q)$ discontinuous w.r.t. $q$ given a fixed $u$ especially at the coverage boundaries induced by the resources' coordinates, for e.g., drones have a circular probabilistic coverage area governed by their height and camera half-angle $\theta$, outside which the coverage probability suddenly drops to zero.

**Theorem 1.** *Let the objective function be as shown in eq 1: $r(u) = \int_Q f(u, q)\, dq$. Denoting the set of points covered by the $i$-th resource as $S_i$, the interior of a set with $in(\cdot)$ and the boundary with $\delta(\cdot)$, the gradient of $r(u)$ w.r.t. the $i$-th resource's location $u_i$ is given by:*

$$\frac{\partial r(u)}{\partial u_i} = \int_{in(Q \cap S_i)} \frac{\partial f(u, q)}{\partial u_i}\, dq + \int_{Q \cap \delta S_i} \left( f(u, q) - f(u_{\setminus i}, q) \right) \frac{\partial q_{Q \cap \delta S_i}}{\partial u_i}^T n_{q_{Q \cap \delta S_i}}\, dq \quad (2)$$

*Proof.* While function $f$ can be potentially discontinuous in $q$ across resources' coverage boundaries, $r(u)$ integrates over $q \in Q$ thereby removing the discontinuities. Hence, instead of directly taking the derivative w.r.t. a particular resource's location $u_i$ inside the integral sign, we split the integral into two parts - over the $i$-th resource's coverage area $S_i$ and outside it:

$$r(u) = \int_{Q \cap S_i} f(u, q)\, dq + \int_{Q \setminus S_i} f(u, q)\, dq \quad (3)$$

Splitting the integral at the boundary of the discontinuity allows us to explicitly capture the effect of a small change in $u_i$ on this boundary. Denoting the interior of a set with $in(\cdot)$ and the boundary with $\delta(\cdot)$, the derivative w.r.t. $u_i$ can be expressed using the Newton-Leibniz formula as:

$$\frac{\partial r(u)}{\partial u_i} = \int_{in(Q \cap S_i)} \frac{\partial f(u, q)}{\partial u_i}\, dq + \int_{\delta(Q \cap S_i)} f(u, q) \frac{\partial q_{\delta(Q \cap S_i)}}{\partial u_i}^T n_{q_{\delta(Q \cap S_i)}}\, dq$$

$$+ \int_{in(Q \setminus S_i)} \frac{\partial f(u_{\setminus i}, q)}{\partial u_i}\, dq + \int_{\delta(Q \setminus S_i)} f(u_{\setminus i}, q) \frac{\partial q_{\delta(Q \setminus S_i)}}{\partial u_i}^T n_{q_{\delta(Q \setminus S_i)}}\, dq, \quad (4)$$

where $\frac{\partial q_{\delta(Q \cap S_i)}}{\partial u_i}$ denotes the boundary velocity for $\delta(Q \cap S_i)$ and $n_{q_{\delta(Q \cap S_i)}}$ denotes the unit-vector normal to a point $q$ on the boundary $\delta(Q \cap S_i)$ (similarly for $\delta(Q \setminus S_i)$). Since $f(u_{\setminus i}, q)$ does not depend on $u_i$, we can set $\frac{\partial f(u_{\setminus i}, q)}{\partial u_i} = 0$. Next observe that the boundaries can be further decomposed as: $\delta(Q \cap S_i) = (\delta Q \cap S_i) \cup (Q \cap \delta S_i)$ and similarly $\delta(Q \setminus S_i) = (\delta Q \setminus S_i) \cup (Q \cap \delta S_i)$. However since $u_i$ does not change the boundary of the target domain $\delta Q$, we have:

$$\frac{\partial q_{\delta Q \cap S_i}}{\partial u_i} = 0, \quad \forall q \in \delta Q \cap S_i \quad (5)$$

$$\frac{\partial q_{\delta Q \setminus S_i}}{\partial u_i} = 0, \quad \forall q \in \delta Q \setminus S_i \quad (6)$$

Further on the boundary of $S_i$, the following unit-vectors normal to the boundary are oppositely aligned:

$$n_{q_{\delta(Q \setminus S_i)}} = -n_{q_{\delta(Q \cap S_i)}} \quad \forall q \in Q \cap \delta S_i. \quad (7)$$

Substituting the above results, we can simplify the gradient expression in eq 4 to:

$$\frac{\partial r(u)}{\partial u_i} = \int_{\text{in}(Q \cap S_i)} \frac{\partial f(u,q)}{\partial u_i} \, dq + \int_{Q \cap \delta S_i} \left( f(u,q) - f(u_{\setminus i}, q) \right) \frac{\partial q_{Q \cap \delta S_i}}{\partial u_i}^T n_{q_{Q \cap \delta S_i}} \, dq \quad (8)$$

$\square$

The first term in eq 2 corresponds to the change in $f$ inside the coverage area of resource $i$ due to a small change in $u_i$, while the second term elegantly factors-in the effects of movement or shape change of the coverage area boundary due to changes in $u_i$ (e.g. when a drone moves or elevates in height). While we show the general result here, the term $\frac{\partial q_{Q \cap \delta S_i}}{\partial u_i}^T n_{q_{Q \cap \delta S_i}}$ can be simplified further using implicit differentiation of the boundary of $S_i$, which depends on the particular game under consideration. We show the simplification for our example domains in section A.2 in the appendix.

### 3.2 DISCRETIZATION BASED APPROXIMATIONS

While we now have a general form for $r(u)$ and $\frac{\partial r}{\partial u}$, both forms comprise of non closed-form integrals over the target domain $Q$ or its subsets. While evaluating $r$ and $\frac{\partial r}{\partial u}$ in practice, we adopt a discretization based approach to approximate the integrals. Given a target domain $Q \subset \mathbb{R}^d$ with $d \in \{2, 3\}$, we discretize the full $\mathbb{R}^d$ space into $B_1, \ldots, B_d$ bins respectively in each of the $d$ dimensions. **Approximating spatial maps**: All spatial maps i.e. functions over the target domain $Q$ (e.g. $f(u,q)$), are internally represented as *real tensors* of dimension $d$ with size: $(B_1, \ldots, B_d)$. **Approximating sets**: All geometric shapes (or sets of points) including $S_i$ for all resources (e.g., the circular coverage areas of drones and lumberjacks) and the target domain $Q$ itself (e.g., the irregular shaped forest) are converted to *binary tensors* each of dimension $d+1$ with size: $(B_1, \ldots, B_d, 3)$. The final dimension of length 3 denotes interior, boundary and exterior of the geometric shape respectively, i.e. a binary tensor $T$ has $T_{b_1, \ldots, b_d, 0} = 1$ if the bin at index $(b_1, \ldots, b_d)$ is inside the geometric shape, $T_{b_1, \ldots, b_d, 1} = 1$ if the bin is on the boundary of the geometric shape and $T_{b_1, \ldots, b_d, 2} = 1$ if the bin is outside the geometric shape. **Approximating operators**: Doing the above discretization requires an efficient function for computing the *binary tensors* associated with the $\text{in}(\cdot)$ and the $\delta(\cdot)$ operators. This is performed by our efficient divide-and-conquer shape discretizer, which is presented in section A.3 due to space constraints. The other set operations are approximated as follows: (a) set intersections are performed by element-wise *binary tensor* products, (b) integrals of spatial maps over geometric sets are approximated by multiplying (i.e. masking) the *real tensor* corresponding to the spatial map with the *binary tensor* corresponding to the geometric set followed by an across-dimension sum over the appropriate set of axes.

While our discretized bins growing exponentially with dimension $d$ of the target domain may come off as a limitation, our method still scales well for most real-world coverage problems since they reside on two or three-dimensional target domains. Further, unlike previous methods which discretize the target domain and simultaneously restrict the agents' actions to discrete bins (Yang et al., 2014; Haskell et al., 2014), we do not discretize the actions $u$ of agents. Hence, we do not run into intractability induced by discretizing high-dimensional actions of agents owning multiple resources and we keep $u$ amenable to gradient-based optimization. Our proposed framework acts as an autograd module for $r(u)$, differentiable w.r.t. input $u$, and provides both the forward and the backward calls (i.e. evaluation and gradients).

## 4 EXPERIMENTS

In our experiments on both our application domains, we differentiably approximate rewards using the following variants: (a) feedforward neural networks [*nn*], (b) graph neural networks [*gnn*], and (c) and with our differentiable coverage approximation [*diff*]. For the *nn* and *gnn* baselines, we trained neural networks, one per forest and per value of $m$ (and $n$ for two-agent games), to predict the reward of the defender (and attacker in case of two-agent game). The neural networks take as input the action $u_D$ of the defender (and $u_A$ also for two-agent game) and outputs a prediction for the reward $\hat{r}_{D,1p}$ ($\hat{r}_{D,2p}$ and $\hat{r}_{A,2p}$ for two-agent game). Please see section A.6 in appendix for network architectures and hyperparameters. We also represent best responses with the following variants: (a) stochastic best response nets [*brnet*] as originally done by DeepFP, and (b) our deterministic evolutionary population [*popK*] with $K$ being the population size. We use $d = 2$ dimensional forests and discretize them into $B_1 = B_2 = 200$ bins per dimension for a total of $40K$ bins.

## 4.1 RESULTS ON AREAL SURVEILLANCE DOMAIN

We maximized differentiable approximations of $\hat{r}_{D,1p}$ using all three methods: *nn*, *gnn* and *diff* for different values of $m \in \{1, 2, 4\}$ over 5 different forest instances differing in shape and tree density. The maximum true reward $r_{D,1p}$ achieved by the three methods in all cases averaged over all the forest instances is summarized in Table 1. It is clear that *diff* always achieves the maximum true reward. While the difference difference from *nn* and *gnn* is less pronounced for $m = 1$, as the number of agent resources increases beyond 1, the approximation quality of *nn* and *gnn* deteriorates and the difference becomes very significant. This is also reflected in the plots of true reward achieved vs training iterations shown in Figure 2. Since *diff* is an unbiased approximator of the true reward[2], the true reward continues to increase till convergence for *diff*. For *nn* and *gnn*, the true reward increases initially but eventually goes down as the defender action $u_D$ begins to overfit the biased and potentially inaccurate approximations made by *nn* and *gnn*[3]. Figure 3 shows the final locations computed for a randomly chosen forest and with $m = 2$ for all three methods.

Table 1: Maximum reward averaged across forest instances achieved for Areal Surveillance domain.

|  | $m = 1$ | $m = 2$ | $m = 4$ |
|---|---|---|---|
| *diff* (ours) | $\mathbf{9366.03 \pm 657.18}$ | $\mathbf{16091.09 \pm 932.77}$ | $\mathbf{25117.98 \pm 1554.34}$ |
| *nn* | $9293.26 \pm 646.37$ | $14649.32 \pm 1206.60$ | $18962.87 \pm 2018.54$ |
| *gnn* | $9294.47 \pm 664.28$ | $14604.11 \pm 1189.48$ | $19353.93 \pm 2701.81$ |

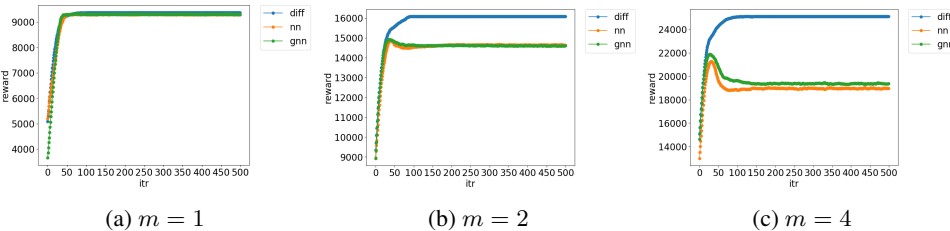

(a) $m = 1$        (b) $m = 2$        (c) $m = 4$

Figure 2: Plots of true reward achieved over DeepFP iterations by *diff*, *nn* and *gnn*.

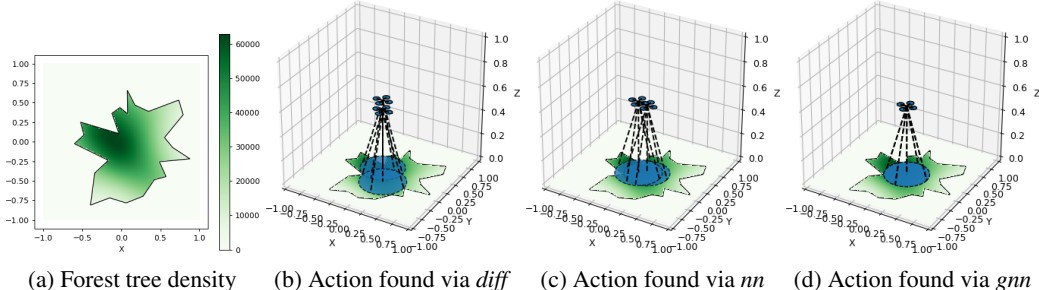

(a) Forest tree density    (b) Action found via *diff*    (c) Action found via *nn*    (d) Action found via *gnn*

Figure 3: Visualizing final actions for a randomly chosen forest with $m = 2$.

## 4.2 RESULTS ON ADVERSARIAL COVERAGE GAME

We implemented different variants of DeepFP with variations of differentiable reward models in $\{nn, gnn, diff\}$ along with variations of best responses in $\{brnet, pop4\}$. We measured the exploitability $\epsilon_D(\sigma_D)$ of the defender strategy found by all methods to compare them against each other. To compute the exploitability of the defender strategy found by any variant of DeepFP, we froze the defender strategy $\sigma_D$ and directly maximized $\mathbb{E}_{u_D \sim \sigma_D}[\hat{r}_A(u_D, u_A)]$ w.r.t. $u_A$ with $\hat{r}_A$ being approximated by *diff*. This is a single-agent objective and can be directly maximized with gradient ascent. We perform 30 independent maximization runs to avoid reporting local maxima and report the best of them as the exploitability. Note that nash equilibrium strategies are the least exploitable strategies, hence the lower the value of $\epsilon_D(\sigma_D)$ found, the closer $\sigma_D$ is to the nash equilibrium strategy.

---

[2]The only bias in *diff* is the discretization bin sizes, which can be made arbitrarily small in principle.
[3]Please see section A.1 in the appendix for a detailed analysis of this phenomenon.

Table 2 shows the exploitability values for different variants of DeepFP. We observe that the exploitability when best responses are approximated by a population-based variant with $K = 4$ is always lower than that of stochastic best response networks employed by original DeepFP. Further, with few agent resources $m = n = 1$, the exploitability across *diff, nn* and *gnn* is nearly similar but the disparity increases for larger number of agent resources and *diff* dominates over *nn* and *gnn* with less exploitable defender strategies. Notably, the original DeepFP (*nn + brnet*) is heavily exploitable while our proposed variant (*diff + popK*) is the least exploitable. In Figure 4, we show a visualization of the points sampled from the defender and attacker's strategies for $m = n = 2$ case on the same forest from Figure 3a. The visualization confirms that *diff + popK* covers the dense core of the forest with the defender's drones so the attacking lumberjacks attack only the regions surrounding the dense core, while *nn + brnet* drones often gets stuck and concentrated in a small region thereby allowing lumberjacks to exploit the remaining dense forest. Please also see section A.5 in the appendix exploring the trade-offs in the choice of population size $K$.

Table 2: Exploitability of the defender averaged across forest instances.

| $\epsilon_D(\sigma_D)$ | $m = n = 1$ | $m = n = 2$ | $m = n = 4$ |
|---|---|---|---|
| | | *brnet* | |
| *diff* (ours) | $209.78 \pm 49.94$ | $399.95 \pm 57.70$ | $559.36 \pm 164.21$ |
| *nn* | $203.92 \pm 54.67$ | $323.00 \pm 39.55$ | $787.53 \pm 194.82$ |
| *gnn* | $204.55 \pm 50.72$ | $307.74 \pm 62.67$ | $597.23 \pm 125.01$ |
| | | *pop4* (ours) | |
| *diff* (ours) | $116.41 \pm 15.02$ | $\mathbf{141.09 \pm 13.90}$ | $\mathbf{141.54 \pm 26.60}$ |
| *nn* | $\mathbf{113.61 \pm 6.92}$ | $208.23 \pm 22.76$ | $339.31 \pm 116.77$ |
| *gnn* | $113.99 \pm 13.74$ | $176.25 \pm 15.21$ | $172.30 \pm 34.08$ |

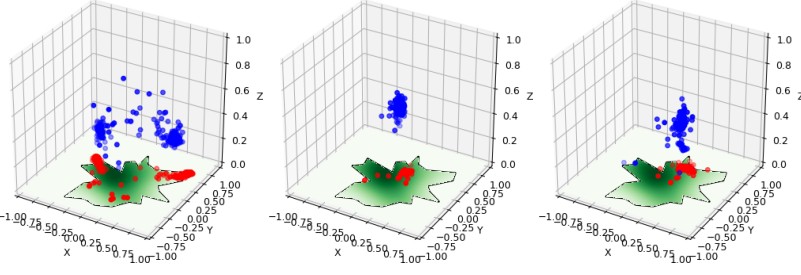

(a) Strategy for *diff + brnet*  (b) Strategy for *nn + brnet*  (c) Strategy for *gnn + brnet*

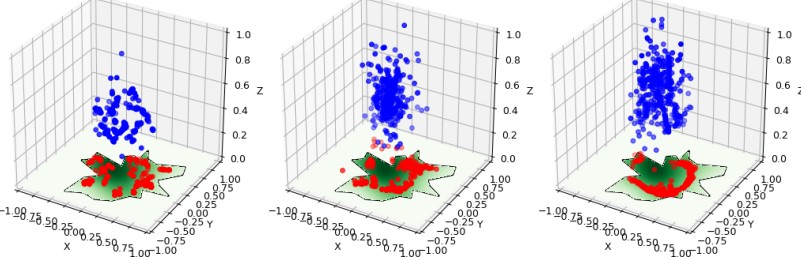

(d) Strategy for *diff + pop4*  (e) Strategy for *nn + pop4*  (f) Strategy for *gnn + pop4*

Figure 4: Visualizing final strategies found via *diff, nn* and *gnn* with best responses of the form *brnet* and *pop4* on a randomly chosen forest with $m = n = 2$. The blue (red) dots are sampled from the defender's (attacker's) strategy for the 2 drones (lumberjacks).

## 5 CONCLUSION

In this work, we show that spatial coverage objectives with multiple-resources are combinatorially hard to approximate with neural networks. We propose to directly approximate a large class of multi-agent multi-resource spatial coverage objectives and their gradients tractably without learning neural network based reward models. By augmenting existing approaches with our spatial discretization based approximation framework, we show improved performance in both single-agent and adversarial two-agent multi-resource spatial coverage problems.

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
