# OpenReview forum: "Differentiable Approximations for Multi-resource Spatial Coverage Problems"
_ICLR.cc/2021/Conference — Reject_

### Official Review · AnonReviewer3 · 2020-10-20
**Very hard to follow paper**

**Rating:** 6
**Confidence:** 1

**Review:**

This paper shows that spatial coverage objectives with multiple-resources are combinatorially hard to approximate with neural networks and proposes a spatial discretization based approximation framework to solve this problem.

The paper is very hard to follow due to all the notations the authors introduced and the details set up in the examples. Many details in the two examples look a little bit extra, which could be properly abstracted.

---

> ### Author Response · Authors · 2020-11-24
> **Thank you for the review**
>
> We understand that you found the paper hard to follow and we have explained the reasons and measures to mitigate it in our joint response to all reviewers posted separately. We will address the writing concerns in an updated version of the paper.

---

> ### Comment · AnonReviewer4 · 2020-11-25
> **Unfair Review**
>
> As a peer reviewer, I feel it is very irresponsible to leave a review like this, and an author's hard work should not be treated like this.

---

### Official Review · AnonReviewer1 · 2020-10-26
**The paper presents a viable method but overall does not meet the standard of ICLR.**

**Rating:** 4
**Confidence:** 4

**Review:**

##########################################################################
Summary:
This paper studies the coverage game where agents allocate their resources to target spaces to maximize their coverage, and the goal of this paper is to (approximately) compute the Nash Equilibrium. The proposed method simulates the game by iteratively updating the best response, and the main contribution is an algorithm to approximate the gradient of the utility function with respect to the resource allocation (over the space). In particular, the paper proposes to decompose the gradient into two parts and estimate each part by discretization.

##########################################################################
While the paper presents a viable method, I vote for reject for several reasons. The paper does not successfully justify the merits of the proposed method, with many hidden parts that are hard to follow. The paper lacks a good organization, and the writing could have been more formal (see detailed comments). The experiments are not convincing.
##########################################################################

Detailed comments:

First of all, I am slightly concerned that this paper's work does not really fit ICLR, as its goal is to approximate the Nash Equilibrium of a game, without any learning process involved. The only part I see relevant is that two learning-based methods were adopted as competitors.

The presentation of this paper lacks clarity and a good structure. I could not figure out the problem this paper wants to solve after reading the first three pages. The abstract and the introduction claim that the paper wants to solve the resource allocation problem, but it turns out the specific target is to approximate the gradient.

From my view, using the proposed decomposition is a doable method, but I cannot see the novelty in doing so. The paper mentions some hardness to be overcome: (a) the integration has no closed-form and (b) the coverage drops to zero outside the coverage area. However, the paper does not justify how the aimed hardness was addressed by the proposed method.

The paper mentions some existing methods (such as DeepPF in a published paper) but does not take them as competitors. I am also confused that the paper names the proposed method also as DeepPF, so how does it compare to the existing DeepPF?

It looks like the competitors in experiments are baselines without much calibration. The information in the appendix does not provide sufficient details. In addition, instead of providing code in the appendix, it could better to include source files and datasets in the supplementary materials. The currently provided information does not support reproducibility.


The organization and writing of this paper could be improved – e.g.,
-	Please specify the domain when introducing a variable.
-	The definition U=\times_{p \in P} {U_p} is problematic – {U_p} is wired since U_p is already a set.
-	The definition cvg: q \times u -> R is problematic – it should be cvg: Q \times u -> R
-	Instead of saying the agents aim to achieve Nash equilibrium, it would be better to say the game will reach the Nash equilibrium. In addition, is the Nash equilibrium unique?
-	“for e.g.” is not correct.

---

> ### Author Response · Authors · 2020-11-24
> **Thank you for the comments**
>
> We thank you for your constructive comments. We first request you to read our joint response to all reviewers posted separately and we further address your individual comments below.
>
> Q1. "I am slightly concerned that this paper's work does not really fit ICLR, as its goal is to approximate the Nash Equilibrium of a game, without any learning process involved."
>
> A1. This paper is not about approximating Nash Equilibria of a game. Please also see our joint response to all reviewers for more clarification on this. As for having no learning involved, our key contribution is to derive and implement an unbiased gradient estimator for coverage problems. In so far as we understand, ICLR also invites papers involving statistical techniques, optimization, gradient estimation and other ML applications. Our work fits as a gradient estimation technique and can also be further extended for general purpose learning involving coverage.
>
> Q2. "The presentation of this paper lacks clarity and a good structure. I could not figure out the problem this paper wants to solve after reading the first three pages. The abstract and the introduction claim that the paper wants to solve the resource allocation problem, but it turns out the specific target is to approximate the gradient."
>
> A2. We understand the confusion caused due to the structure of the paper. We will address this in the updated draft. Please also see our joint response to all reviewers.
>
> Q3. "From my view, using the proposed decomposition is a doable method, but I cannot see the novelty in doing so. The paper mentions some hardness to be overcome: (a) the integration has no closed-form and (b) the coverage drops to zero outside the coverage area. However, the paper does not justify how the aimed hardness was addressed by the proposed method."
>
> A3. As you rightly mentioned, the hardness is in dealing with the finite coverage fields and not having a closed-form for the integrals. The former, namely the discontinuity in coverage has been addressed in the proof of Theorem 1 (sec 3.1). The key idea is to break the integral in two parts at the resource's coverage boundaries and then apply the Newton-Leibniz theorem to both parts separately. We subsequently use geometric relations between the boundaries of resulting integrals and implicit boundary differentiation to derive the full gradient expression. The latter, namely, the approximation of integrals is handled using spatial discretization and is explained in section 3.2.
>
> Q4. "The paper mentions some existing methods (such as DeepPF in a published paper) but does not take them as competitors. I am also confused that the paper names the proposed method also as DeepPF, so how does it compare to the existing DeepPF?"
>
> A4. We have included both results of the original DeepFP and of our modifications to it in table 2. We still call the method DeepFP since we have not introduced a fundamentally new game theoretic algorithm in this paper (as also mentioned in our joint response to all reviewers), but rather replaced certain sub-components of DeepFP with our contributions to highlight their effectiveness. The numbers for original DeepFP were in the "brnet" part of Table 2 with the nn-function approximator (also mentioned in sec 4.2 on page 8).
>
> Q5. "It looks like the competitors in experiments are baselines without much calibration. The information in the appendix does not provide sufficient details."
>
> A5. We have tried to adequately tune, calibrate and even replace parts of DeepFP wherever applicable and have provided results with all the variants in Table 2. If you can let us know exactly what is missing, we will be happy to provide the details and/or calibrate our approach better.
>
> Q6. "Instead of providing code in the appendix, it could better to include source files and datasets in the supplementary materials. The currently provided information does not support reproducibility."
>
> A6. The pseudo-code in the appendix is not for reproducing the complete project. The full code-base is large and would indeed have to be uploaded in a separate repository. The pseudo-code is only present to explain a sub-component of the framework, namely, our efficient divide-and-conquer discretizer to discretize sets into binary tensors. We will be happy to upload the full code as a separate repository upon acceptance.
>
> Q7. Other writing improvements.
>
> A7. We are happy to incorporate the other writing changes that you have suggested in the updated draft.
>
> We hope that the above explanations clarify your concerns.

---

### Official Review · AnonReviewer2 · 2020-10-27
**A weak accept**

**Rating:** 5
**Confidence:** 2

**Review:**

The main goal of this paper is to address the problem of multi resource spatial coverage.
The main challenges are the non differentiability of the utility function and the complexity of the action space making this problem hard to optimize.
The authors make a clear review of the state of the art approaches that often focus discretizing the action space. In the contrary, the authors propose to discretize the target space to construct a differentiable approximation of the utility function.

I'm really not an expert on multi resource spatial coverage, so I am not able to judge if the presented results are original, nor significant.

However, I found the paper clearly written, and the subject nicely presented with examples.

I suggest a few points of improvement:
- Part of the conclusion and the abstract mention that the paper proved that "multiple-ressources are combinatorially hard to approximate with neural networks". I didn't see proofs of that statement, I agree that the authors develop an approach that doesn't seem to suit NN and they outperform a few NN. This can be enough to show that the presented method is efficient but it is not enough to claim that the compared methods are not.
- I read appendix A1, and I still do not understand why the NN curves, why do they have biases ?
- In table 1 and 2, it would be great to indicate how confidence intervals are computed.


----- Edit after rebutal ------

Thank you for the answers.
The authors only answered partly my concerns. I'm still not convinced by their explanation on the NN curves. I don't understand how their performance decreases with the number of iterations, it seems that they weren't well tuned.

Anyway I've been convinced by R4 and R1 that a few baseline models are missing.

---

> ### Author Response · Authors · 2020-11-24
> **Thank you for the review**
>
> We thank you for your constructive comments. We first request you to read our joint response to all reviewers posted separately and we further address your individual comments below.
>
> Q1. "I read appendix A1, and I still do not understand why the NN curves, why do they have biases ?"
>
> A1. The NNs have biases because they are continuous function approximators trying to approximate the coverage reward which is intrinsically formed by combination of resources with finite coverage fields which have discontinuities at their coverage boundaries.
>
> Q2. "In table 1 and 2, it would be great to indicate how confidence intervals are computed.""
>
> A2. Tables 1 and 2 present their respective metrics with averages and standard deviations over 5 different forest instances (also described in the first para of sec 4.1).
>
> We hope that the above explanations clarify your concerns.

---

### Official Review · AnonReviewer4 · 2020-10-28
**An OK but not good submission**

**Rating:** 4
**Confidence:** 4

**Review:**

##########################################################################
Summary:

This paper studies the (multi-)agent spatial coverage problem. They propose a framework that can approximate the general class of spatial coverage objectives and their gradients via spatial discretization methods.

##########################################################################
Reasons for score:
The contribution of this paper seems trivial, it is a refinement of training DeepFP [Kamra 2019] by deriving a differentiable loss function. The comparisons to existing baselines are missing. I therefore recommend for rejection.


##########################################################################
Pros:

1. This paper studies a specific problem of resource allocation problem in a continuous setting. I appreciate two detailed examples of problem formulation for both single-agent and multi-agent settings.

2. The main contribution of this paper is the loss function derivative in Theorem 1, which mainly uses classical calculus.

##########################################################################
Cons:

1.	The author believes discretization the continuous domain is not a good way to solve resource allocation problem. However, the proposed method still have to adopte the discretization on computing the integral of r and \partial r/u. Although the author emphasise that they do not need to discretize the action domain, but it seems to me that why it is significant is unclear.

2. The author mentioned that model-free methods such as PSRO, has the issue of low sample efficiency and high variance issue for policy gradient methods, however, both of which has not been addressed by this work, and this proposed method is not compared to model-free method either. PSRO and its variants, for example, have shown many success on blotto type of game, so I think the advantages over PSRO [Lanctot 2017] type of methods need clarifying. I would consider improving the score if author can show comparative advantage over PSRO.

---

> ### Author Response · Authors · 2020-11-24
> **Thank you for your comments**
>
> We thank you for your constructive comments. We first request you to read our joint response to all reviewers posted separately and we further address your individual comments below.
>
> Q1. "It is a refinement of training DeepFP [Kamra 2019] by deriving a differentiable loss function."
>
> A1. Please see our joint response to all reviewers.
>
> Q2. "The author believes discretization the continuous domain is not a good way to solve resource allocation problem. However, the proposed method still have to adopte the discretization on computing the integral of r and \partial r/u."
>
> A2. Saying that discretization is not a good way without context can be misleading! Our claim is that when one discretizes the action set for placing the resources, it results in a combinatorial explosion in the solution space. For instance, discretizing the (x,y,z) coordinates of drones in our example 1 into B bins along each axis gives B^3 locations for the placement of a single drone. With m drones (a.k.a. resources), one has to test for B^(3m) locations which scales exponentially with m. With just 100 bins per axis and just one drone, one already has to explore 100^(3) = 1 million bins. For as small as m=2, the number becomes 10^12!! Instead, we keep the action space U undiscretized so we can exploit the continuity in action u and incrementally tune it using gradients (\partial r / \partial u). We only discretize the underlying target domain (not the same as action space) into bins when approximating the spatial integrals required during evaluation of r and its above-mentioned gradient. Since its size is fixed and independent of m, such discretization does not lead to any combinatorial explosion with increasing number of resources (m).
>
> Q3. Comparison against PSRO.
>
> A3. We do not introduce any new game-theoretic algorithms in this paper to try and outperform existing methods like PSRO (as also described in our joint response to all reviewers). The mention of sample efficiency and high variance of policy gradients is to motivate the usage of actor-critic methods in the subsequent paragraph of the introduction. This paper does not aim to address sample efficiency and high variance of policy gradient based methods. We will rectify the statement appropriately in the next draft to avoid any future confusions about our contributions.
>
> We hope that the above explanations clarify your concerns.

---

> > ### Comment · AnonReviewer4 · 2020-11-25
> > **not convinced at all**
> >
> > I read the rebuttal, and feel none of my questions has been addressed. I will maintain my rejection decision.

---

### Author Response · Authors · 2020-11-24
**Joint response to all reviewers**

We thank all the reviewers for their valuable feedback and constructive comments. We want to clarify some misunderstandings about the paper for all reviewers and will address the individual queries from all reviewers in separate responses.

This paper's key contribution is to derive an unbiased gradient estimator for the multi-resource coverage problem and provide a discretization-based framework to approximate it. In this process, we address the following key challenges: (a) each resource has discontinuities at the boundaries of its coverage field, and (b) the objective function takes the form of an integral over an arbitrary [and potentially non-convex] shaped target domain, which has no closed-form expression.

We then show the application of our gradient estimation framework for both single and two agent multi-resource coverage problems. However, due to the inclusion of the adversarial two-agent domain as a potential application, we have had to include considerable discussion on game theory, nash equilibria and fictitious play based methods while formulating the problem and solution techniques. We have also had to dedicate a sizeable chunk of space to experiments involving variants of the DeepFP method (Kamra et al. 2019). This has led to the reviewers:
1. Having trouble understanding the contributions [R1],
2. Finding the notation cumbersome [R3],
3. Believing that the paper is about approximating Nash Equilibrium [R1], and
4. Feeling that our method is just an incremental improvement over DeepFP [R4] and whether it can outperform other game theoretic methods like PSRO [R4].

We want to clarify that our work is not focused on developing a new method to approximate Nash Equilibria or compete against existing methods like DeepFP or PSRO. Our work proposes an unbiased gradient estimation framework for coverage problems and to showcase its use-cases we have applied our framework as a sub-component within existing methods: (a) within gradient ascent for the single agent coverage example, and (b) within DeepFP for the adversarial two-agent example. The novelty of our work lies in proposing the gradient estimation framework for such applications, where it would normally not be possible to use gradient-based methods since an unbiased gradient estimator is unavailable or otherwise one would have to train differentiable approximators, e.g. neural nets, to get an estimate of the gradient (like DeepFP does).

We plan to substantially re-write the paper for the next draft to make the notation less cumbersome and move the game-theoretic aspects into a single section so that the true contributions of the paper are better highlighted.

---

### Decision · Program_Chairs · 2021-01-07
**Final Decision**

**Decision:**

Reject

**Comment:**

The reviewers agreed that there were a few issues with the current version of this work, mainly:

- Some missing baselines that are mentioned in the paper, but not sufficiently compared to

- Problems with the presentation that did not make it easy to understand.

- Not an optimal fit with the intended audience of this conference.